# Deformation Detection in Cyclic Landslides Prior to Their Reactivation Using Two-Pass Satellite Interferometry

Pierpaolo Ciuffi [1], Benedikt Bayer [2], Matteo Berti [1], Silvia Franceschini [2] and Alessandro Simoni [1,*]

[1] Department of Biological, Geological and Environmental Sciences, University of Bologna, Via Zamboni 67, 40126 Bologna, Italy; pierpaolo.ciuffi2@unibo.it (P.C.); matteo.berti@unibo.it (M.B.)

[2] Fragile s.r.l., Viale Fanin 48, 40127 Bologna, Italy; benedikt.bayer86@gmail.com (B.B.); silvia@fragilesrl.it (S.F.)

[*] Correspondence: alessandro.simoni@unibo.it

**Abstract:** Landslides are widespread geological features in Italy's Northern Apennines, with slow-moving earthflows among the most common types. They develop in fine-grained rocks and are subject to periodic rainfall-induced reactivations alternating to phases of dormancy. In this paper, we use radar interferometry (InSAR) and information about landslide activity to investigate deformation signals on an areal basis and to assess the dynamics of recently reactivated earthflows. We use traditional two-pass interferometry by taking advantage of the short revisit time of the Sentinel 1 satellite to characterize 4 years of slope deformations over the 60 km$^2$ study area, where 186 landslides are mapped. Our results show that most intense and sustained deformation signals are associated with phenomena on the verge of reactivation, indicating that radar interferometry may have a potential for early warning purposes. By focusing on three specific earthflow reactivations, we analyze their dynamics through the years that preceded their failure. Despite inherent uncertainties, it was possible to retrieve the deformation signal's temporal evolution, which displayed seasonally recurring accelerations, peaking during the major precipitation episodes in the area.

**Keywords:** InSAR; landslides; earthflows; reactivation; rainfall-induced landslides

## 1. Introduction

Landslides represent one of the most diffuse and problematic natural hazards in many mountain chains of the world [1]. In many countries, landslides are responsible for large economic losses, damages to houses and infrastructures and fatalities [1,2]. They are common morphological features throughout the whole Northern Apennines of Italy.

Here, most slope deformations occur on old landslide bodies that already failed in the past [3] and continue to exhibit repeated partial or complete reactivations. In many cases, the reactivation of old deposits causes the regression of the main scarp and the physical degradation of the material, which moves downwards as an earthflow. In other cases, the reactivation is more complex and different types of landslides may be present on the same slope [4]. In this context, accurate landslide mapping and detection are important for a correct hazard assessment.

Conventional methods used for mapping and monitoring slope instabilities could benefit from remote sensing systems, which allow rapid and easily updatable acquisitions of data over wide areas, reducing fieldwork and costs [5–7].

A powerful technique for the displacements monitoring of large areas is the synthetic aperture radar interferometry (InSAR, DInSAR) that can measure the deformations of the landslide deposits during the slow-motion stage [8].

Radar interferometry has been successfully used to assess subsidence [9], volcanic inflation or deflation [10], and the deformation field of earthquakes [11,12]. It was also applied to detect the deformation caused by landslides [13,14] as well as subsurface excavation and tunneling [15].

The first application of radar interferometry to a landslide investigation dates back to the mid-1990s [16]. However, it was only in the following decade that this technique attracted the attention of the landslide community. Recent advances in radar satellite capabilities (e.g., high spatial resolution and high temporal frequency acquisitions), together with the development of new robust techniques based on the interferometric analysis of large datasets of radar images (multi-interferogram approach), such as the permanent scatterers (PS) [17] or the small baseline technique (SBAS) [18], have increased the potential of remote sensing for landslide investigations. One of the main limitations of multitemporal techniques is the absence of stable reflectors on most active landslides, making them difficult to apply in many critical cases. Traditional two-pass InSAR allows the detection of slow-moving landslides up to several cm per year, following the classifications of Cruden and Varnes, 1996 [19], and was successfully used to investigate the seasonal kinematics of earthflows in California [14–20]. However, before 2015 only L-band data delivered spatially quasi-continuous data in settings like the Northern Apennines. Different InSAR techniques were used in the past to retrieve spatial and temporal displacement of slow slope deformations [13,20–22]. The launch of the new Sentinel 1 satellite constellation, which is characterized by a high acquisition frequency of up to six days, is suited to reduce decorrelation in the derived interferograms [23,24] and permits obtaining promising results with higher temporal resolution [25].

This paper investigates the temporal evolution of surface displacements in a mountainous area characterized by widespread landslides using standard two-pass InSAR analysis. The study area is located in the Northern Apennines of Italy, has an extent of 60 km$^2$, and during the 4 years of analysis (from January 2016 to December 2019), was subject to the reactivation of three existing landslides caused by intense rainfall.

We used an areal InSAR analysis to detect active slope movements during the period of interest. Then we used a local analysis to investigate the relationship between rainfall and landslides acceleration events. The results show that this technique can retrieve spatially quasi-continuous deformation maps, even in areas characterized by the absence of good quality reflectors (mainly buildings and infrastructures). Moreover, through a careful interferogram analysis and selection, it is possible to characterize the evolution of landslide displacement through time and its relationship with the rainfall regime.

## 2. Study Area

The study area has an extension of 60 km$^2$ and is located in the Reno River valley South of Bologna in Italy's Northern Apennines. The Northern Apennines is a collisional belt formed from the convergence of the European and Adriatic plates between the Cretaceous and Early Miocene [26,27]. In the study area, the most common lithologies are chaotic clay-shales with block-in-matrix fabric [28,29] and deposits of turbidity currents, also known as flysch [30]. Flysch consists of a stratified alternation of sandstone, siltstone, and marlstone beds. In the following, we use the term "coarse-grained flysch" and "pelitic flysch" to indicate heterogeneous rocks where fine-grained beds are, respectively, less or more abundant than coarse-grained beds. Landslide types are related to bedrock lithology: earthflows dominate in the clay-shales [31], while a broader spectrum of landslide types can be observed in the flysch. However, the most common phenomena consist of translational or rotational earth- or rockslides in the upper part of the slopes that propagate as earthflow further down-slope [32–34].

This style of activity is typical for the whole Emilia-Romagna Region. Bertolini and Pellegrini, 2001 [4] reported more than 32,000 landslide bodies in the regional territory, most of which can be described as complex landslides, associating roto- transitional slides with earthflows. According to the authors, typical velocities are millimeters to centimeters per year during the dormant phase (which may last years to hundreds of years) and increase up to meters per hour during periodic reactivations. These rapid stages of movement typically occur after periods of heavy rainfall and can lead to the transition of the moving mass from slide to flow [35].

Within the area of interest (Figure 1), the regional inventory reports the presence of 186 landslides that were classified by the local Geological Survey as "active" or "dormant" based on geomorphic evidence. Approximately 18.8% of them are set within coarse-grained flysch, 31.2% within pelitic flysch, 38.2% within chaotic clay shales, and only 11.8% within other sedimentary rocks. Only four landslides underwent paroxysmal reactivation during the analysis period (2015–2019). All of them are located inside the chaotic clay shales. One of these landslides (the Marano earthflow) occurred in March 2020 and has already been studied using radar interferometry [8]. The other three (Braina, Carbona, and Spareda) occurred in November–December 2019 and are the subject of this study.

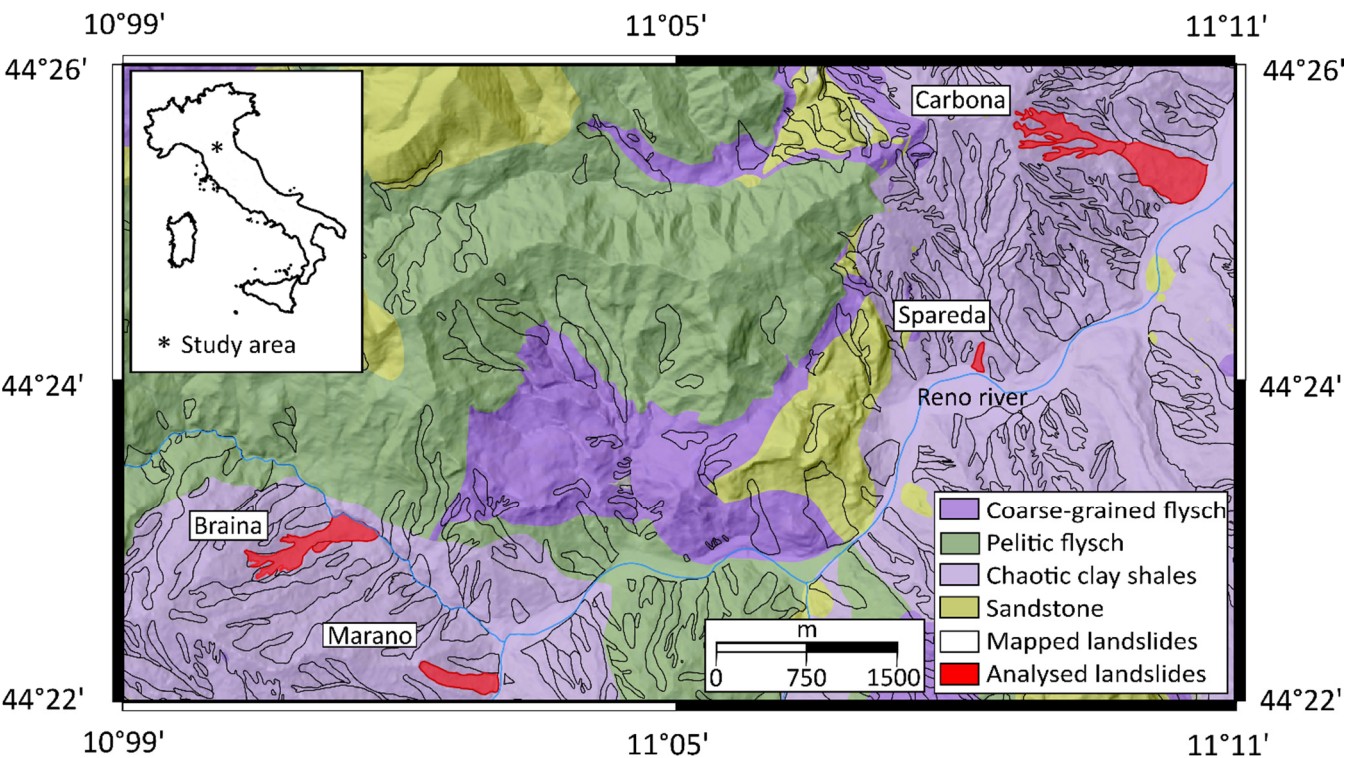

**Figure 1.** Schematic map with elements of the geology of the study area; labeled landslides are those reactivated during the study period (Geological, Seismic and Soil Service of the Emilia Romagna Region).

The area has a Mediterranean climate, with total annual precipitation of about 1300–1400 mm [36]. The temporal precipitation pattern is characterized by intense rainfall in spring and autumn, separated by dry summers and relatively dry winter months when precipitation partly occurs as snowfall [37–39].

### 2.1. The Braina Landslide

The Braina landslide (Figure 2) is located within the municipality of Gaggio Montano, in the province of Bologna (Italy), between 375 and 596 m a.s.l. It is 1250 m long and between 100 and 400 m wide, with an average slope of 12.6 degrees. The landslide reactivated as an earthflow [19] inside chaotic clay-shales belonging to the Ligurian domain, known as Palombini Shale Formation [40].

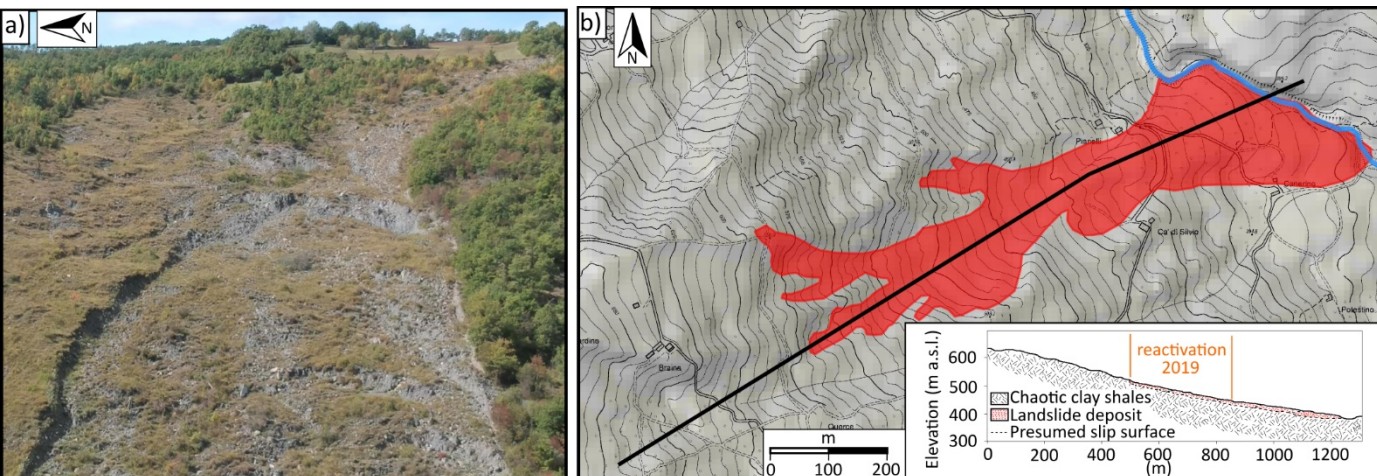

**Figure 2.** UAV (Unmanned Aerial Vehicle) photograph of Braina earthflow taken after reactivation (**a**). Schematic map and profile of the landslide deposit (**b**).

Since 1900, the landslide reactivated three times. The first event took place at the beginning of March 1934 after a prolonged rainfall [41]. Then, the landslide remained dormant for more than 60 years and reactivated a second time on 30 October 1999. According to the eyewitnesses, the landslide activated as a translational movement and then evolved into an earthflow. The reactivated area was 780 m long and between 60 and 17 m wide; the overall extent was about 47,000 m², and the approximate volume 470,000 m³. The landslide disrupted the minor hydrographic network, damaged a building, and destroyed the municipal road. Local authorities carried out extensive works to consolidate the slope, such as draining trenches, hydrographic network arrangement, bridles, and wooden crib walls [41]. Despite these efforts, the landslide reactivated again in late 2019, between 15 and 21 November, due to the continuous precipitation in the second half of November. The reactivated area during this event was 1250 m long and between 100 and 450 m wide. The overall extent was about 23,000 m². This event partially damaged the municipal road and destroyed most of the ditches realized in 1999.

### 2.2. The Carbona Landslide

The Carbona landslide (Figure 3a,b) is located within the municipality of Vergato (province of Bologna, Italy). It develops on a slope composed of chaotic clay-shales belonging to the Palombini Shale Formation [40]. This landslide has been known since the first half of 1800, and, up to the present day, six activations are reported [41]. The first one occurred during the spring of 1837. The second one occurred in late 1901 and caused serious damages to the main road located at the toe of the slope. The third activation dates 28 September 1937. In this case, the reactivated portion had a length of more than 1000 m and an average width of 500 m. In this case, the landslide caused damages, such as the displacement of some meters of telegraph and electricity poles. Another activation occurred during the night between 27 and 28 November 1965, affecting a 900 m long and 150 m wide area. The last historical event took place in the night between 5 and 6 November 1979. On that occasion, the landslide showed signs of movement in the previous months (May and September, according to the reports). The failed mass had a length of 1500 m and an average width of 1000 m for an area of 300,000 m². Rainfall was the cause of the trigger. The landslide damaged a house (declared uninhabitable), forced local authorities to evacuate several families and created swellings and cracks for a stretch of about 100 m along the main road at the toe [41]. The portion that reactivated during the last event during December 2019 develops from an altitude of 308 to 564 m a.s.l. with an average slope of 16.5 degrees. During this last reactivation, the landslide moved as earthflow [19] after the persistent rainfall in mid and late November. Furthermore, In this case, damages occurred to the municipal road that crosses the slope.

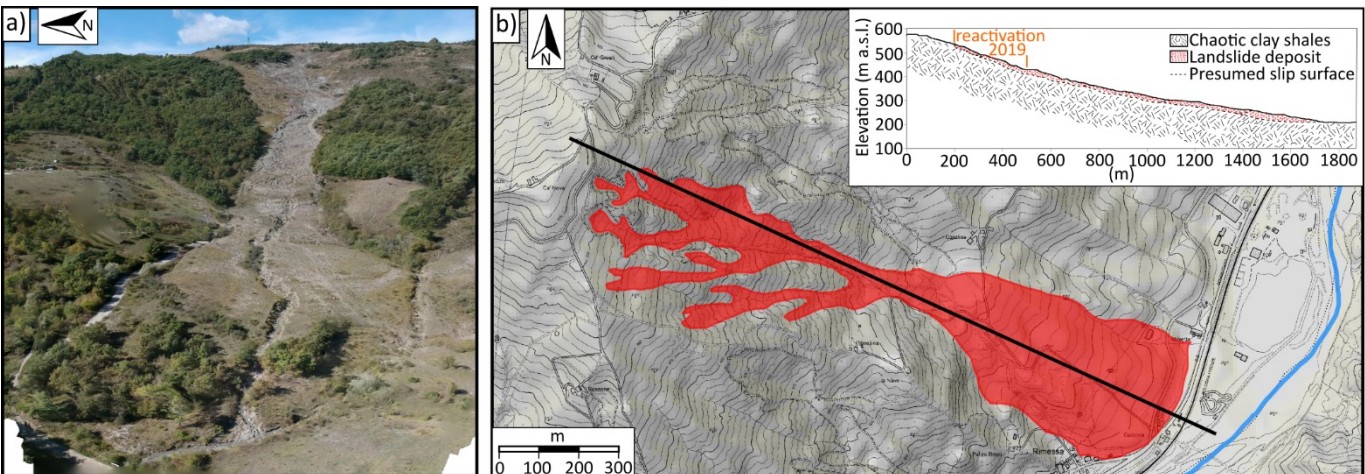

**Figure 3.** UAV photograph of the Carbona earthflow taken after reactivation (**a**). Schematic map and profile of the landslide deposit (**b**).

### 2.3. The Spareda Landslide

The Spareda landslide (Figure 4a,b) is located within the municipality of Vergato (province of Bologna, Italy). It was last reactivated as a rotational slide [19] between 1 and 6 December 2019 due to the persistent rainfall of November.

The landslide develops between an altitude of 230 and 285 m a.s.l.; it has an average slope of 15.7 degrees, length of 1300 m and width between 50 and 150 m, with an area of about 19,000 m². During this last event, the landslide destroyed the municipal road that crosses the slope. Furthermore, this landslide is hosted by the chaotic clay-shales of the Palombini Shales [40]. The regional inventory [41] reports that a previous activation occurred during mid-March 2011. In this case, the extent of the landslide is difficult to determine exactly. Both the 2011 and 2019 reactivations were caused by rainfall, but we cannot exclude that the erosion at the foot of the slope by the Reno River [41] also had a considerable influence. This second cause may also have contributed to the reactivation that occurred during December 2019.

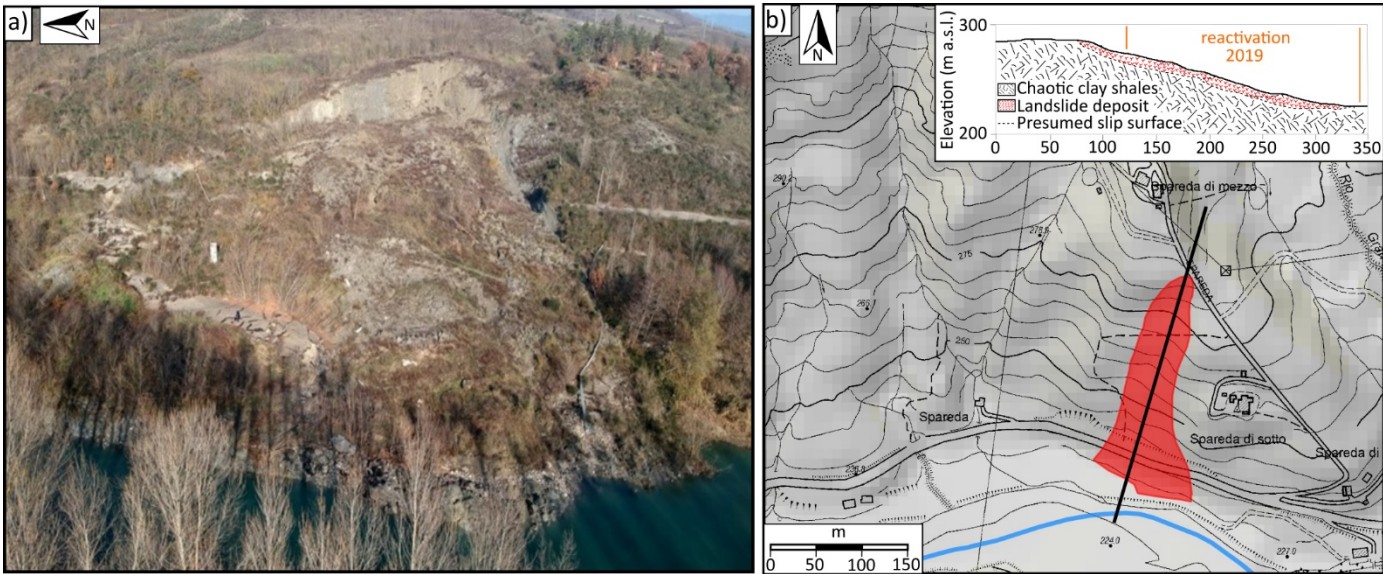

**Figure 4.** UAV photograph of Spareda landslide taken after reactivation (**a**). Schematic map and profile of the landslide deposit (**b**).

## 3. Material and Methods

### 3.1. Synthetic Aperture Radar Interferometry (InSAR): Techniques, Potentials and Limitations

Since the first studies that used radar interferometry to map differential displacements over agricultural fields [42], synthetic aperture radar (SAR) interferometry, or InSAR, has become a useful remote sensing technique to study surface deformations [43]. Different InSAR techniques were used in the past to retrieve spatial and temporal deformation of slow slope deformations [14,20–22].

Data from numerous spaceborne synthetic aperture radar (SAR) missions with different wavelengths are available to retrieve surface deformation by radar interferometry, and the basic principles of this differential remote sensing technique are reviewed for instance, in Rosen et al. 2000 [44], Burgmann et al. 2000 [11], while examples of geological engineering applications can be found in Colesanti and Wasowski 2006 [13]. The major limitations for this technique are the following factors:

1. Phase ambiguity: The differential phase of an interferogram is measured as a fraction of the wavelength, and a deformation field is mapped in the range between $-\pi$ and $\pi$ radians. At this point, the phase is generally named wrapped phase, and in deforming areas, a spatial pattern called interferometric fringes can often be observed [45]. Switching from one end of the spectrum to the other is also commonly called phase-jump. Resolving this phase-ambiguity to obtain absolute values requires a process that is called phase unwrapping and can be solved numerically by different approaches [46,47];

2. Decorrelation: Especially in rural or densely vegetated areas, decorrelation of the interferogram can happen [48]. It is mainly due to surface changes between two acquisitions (temporal decorrelation), which can result from high deformation rates, rapid vegetation growth, snow cover or the long period of the interferogram. Decorrelation may also occur if the distance of the sensors between two acquisitions (known as perpendicular baseline) is large, which is called baseline decorrelation. If coherence is low, unwrapping will also be more problematic [49];

3. Phase noise: Even if the interferometric phase is coherent, it could contain unwanted residual noise due to the differential phase generated by DEM errors, atmospheric phase delay, and orbital inaccuracies [49–51].

Several multitemporal techniques, such as persistent scatterers interferometry [10,17,52], small baseline techniques [9–18], or hybrid approaches [53], were developed to address the problems of decorrelation and estimate different error terms of the phase. More recently, various authors [8,22,25] have shown that traditional two-pass interferometry can be successfully used to investigate gravitational slope movements thanks to shorter satellite revisit intervals and/or improved radar image quality.

### 3.2. InSAR Dataset and Processing

In this work, synthetic aperture radar images captured by Copernicus Sentinel 1 A-B satellites were used. These images are acquired in C-Band (5.6 cm radar wavelength). The acquisition intervals are 12 days before August 2016 and reach 6 days after the launch of Sentinel 1 B. Interferogram processing was done using GMTSAR [53], and interferograms were unwrapped with the statistical-cost network-flow algorithm SNAPHU [46]. We studied the period between January 2015 and December 2019 by analyzing two ascending and two descending orbits (for a total of four datasets for each landslide). The orbits used are the following: track 15 and 117 (ascending); Track 168 and 95 (descending). In general, we obtained better quality results in terms of coherence from tracks 117 and 95, which is why the results described here are obtained from those datasets. We performed an areal InSAR analysis over the whole study area to identify landslides showing deformation signals with a certain temporal continuity. Then, to analyze the extent and time interval of the deformation in more detail, a site-specific analysis was developed for each landslide. The topographic phase was calculated and subtracted [11,45] by using a digital surface model ($2 \times 2$ m$^2$ DSM, provided by the Emilia Romagna Region Services). Large-scale

atmospheric noise was treated and reduced by selecting a stable reference area as close as possible to each investigated landslide. All the stable reference areas used in the different analyses were selected in areas outside the deforming regions, typically w.r.t. anthropic features (e.g., stable buildings) located near the landslides of interest. In addition, Gaussian and Goldstein filters [54] were applied to reduce the noises, eliminate phase differences related to other factors (atmospheric, etc.) and enhance the deformation signals due to landslide displacements. In periods characterized by high displacement rates, unwrapping problems continued to occur despite the acquisition frequency of six days. Only in these cases, we adopted a method proposed by Handwerger et al. 2015 [22], which consists of using a deformation model to aid the unwrapping problem. Both for the areal and the site-specific InSAR analyses, approximately 3000 interferograms were processed and inspected carefully. The inspection is accompanied by a coherence measurement, which is generally satisfactory given the small size of the temporal baseline. Based on our experience, the evaluation of the average coherence is only partially helpful, and the visual inspection of the interferograms is crucial. In fact, interferograms with high coherence values may contain noticeable residual noise and be discarded, while interferograms with low coherence values may contain displacement information that includes them in the selected dataset. Only the interferograms, where residual noise was relatively low and where the phase ambiguity was correctly solved, were used in the final analyses. The temporal baselines of selected interferograms have intervals of 6, 12 and 24 days, while the spatial baselines never exceed 200 m.

## 4. Results

### 4.1. InSAR Areal Analysis

The areal InSAR analysis was performed over the entire study area (60 km$^2$) and covered the period from February 2016 to December 2019. Figure 5 reports an interferometric stack acquired in descending viewing geometry that spans the period 2018–2019. Almost the entire territory shows little or no movement, with displacement rates less than 20 mm/year (green pixels). Extensive decorrelated areas (gray/transparent pixels) are observed along the steep slopes exposed to the North.

Only four slopes show consistent signs of movement during the period of our analysis (Figure 5). Three of them (Braina, Carbona and Spareda) correspond to the landslides that suffered catastrophic failures during November–December 2019; the fourth is represented by the Marano landslide, which was reactivated during March 2018. The detected interferometric signals can be interpreted as slow downslope displacements before the reactivation. In fact, interferograms related to the late 2019 reactivation events are excluded from the areal stacking (Figure 5). Therefore, the analysis refers to the years prior to the catastrophic failure.

The analysis of the temporal evolution of deformation signals, which was developed by observing the monthly areal stacks, reveals that Braina, Carbona and Spareda were actively deforming during the entire analysis period, preceding their reactivation. The Marano landslide reactivated catastrophically in March 2018, and the deformation signal is less evident (i.e., lower average displacement rate). In this case, the deformation takes place mostly during the first months of 2018 [8], while in the post-failure phase, only small displacements are observed. According to the regional landslide database, no other landslides were exhibited high displacement rates in the study area during the observation period. Therefore, no landslides went undetected.

These results show that the InSAR analysis can provide useful information on the landslide activity for most of the mountain territory (~85%) despite the scattered and sparse presence of infrastructures. The only areas without information are the decorrelated slopes characterized by dense vegetation on steep slopes exposed to the North.

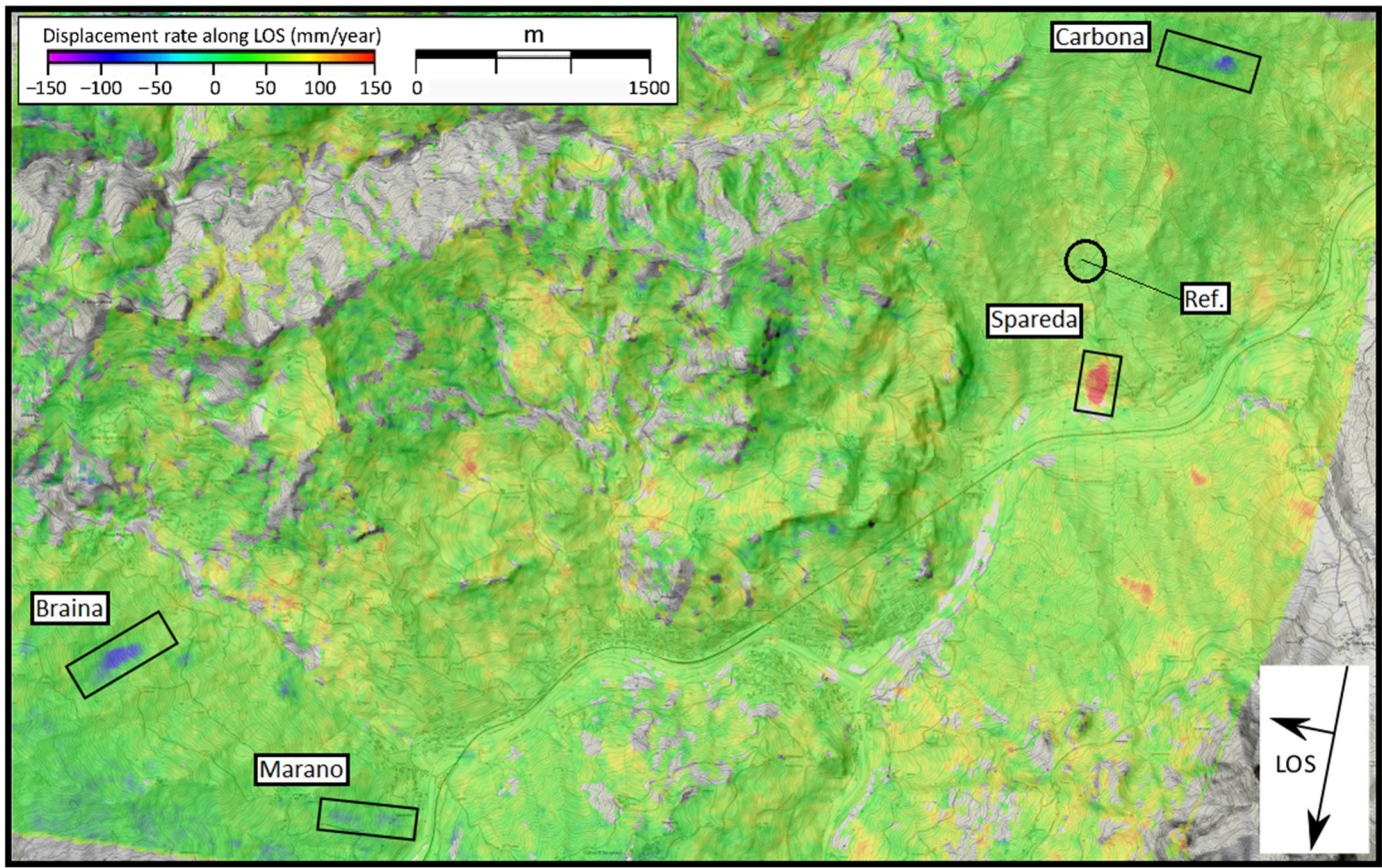

**Figure 5.** Results of areal radar interferometry (InSAR) analysis (descending orbit #95), 2018–2019 interferometric stack.

### 4.2. InSAR Site-Specific Analysis

A site-specific analysis was carried out to obtain more detailed information about the kinematics of the three landslides detected by the areal analysis. Besides using different reference areas, we performed an independent interferogram selection for each case study. The results described below consist of deformation maps obtained by stacking the selected interferograms of the whole analysis period (2016–2019). Figure 6 shows the total stacks obtained for the ascending and descending orbits for the three case studies. The deformation fields show similar shapes in both orbits, while the direction of the displacement is inverted for the different orbits. This difference is common in landslides and should be interpreted as a displacement along the downslope direction.

As expected, in the case of Braina and Carbona, the analysis reveals positive displacement rates (movement away from the satellite) for the ascending orbit and negative displacement rates for the descending orbit along the east-facing slopes.

In the case of Spareda, the displacement signals are reverted in relation to the orbit and compatible with the south-west facing slope. The total stacks indicate that the three landslide bodies have been subject to some degree of deformation in the four years preceding the catastrophic failure.

The Braina earthflow (Figure 6a,b) has been subject to pre-failure deformations mostly affecting the upper part, between approximately 440 m and 550 m a.s.l. This area shows average values of LOS(Line Of Sight)-displacements ranging from 40 mm/year to 120 mm/year.

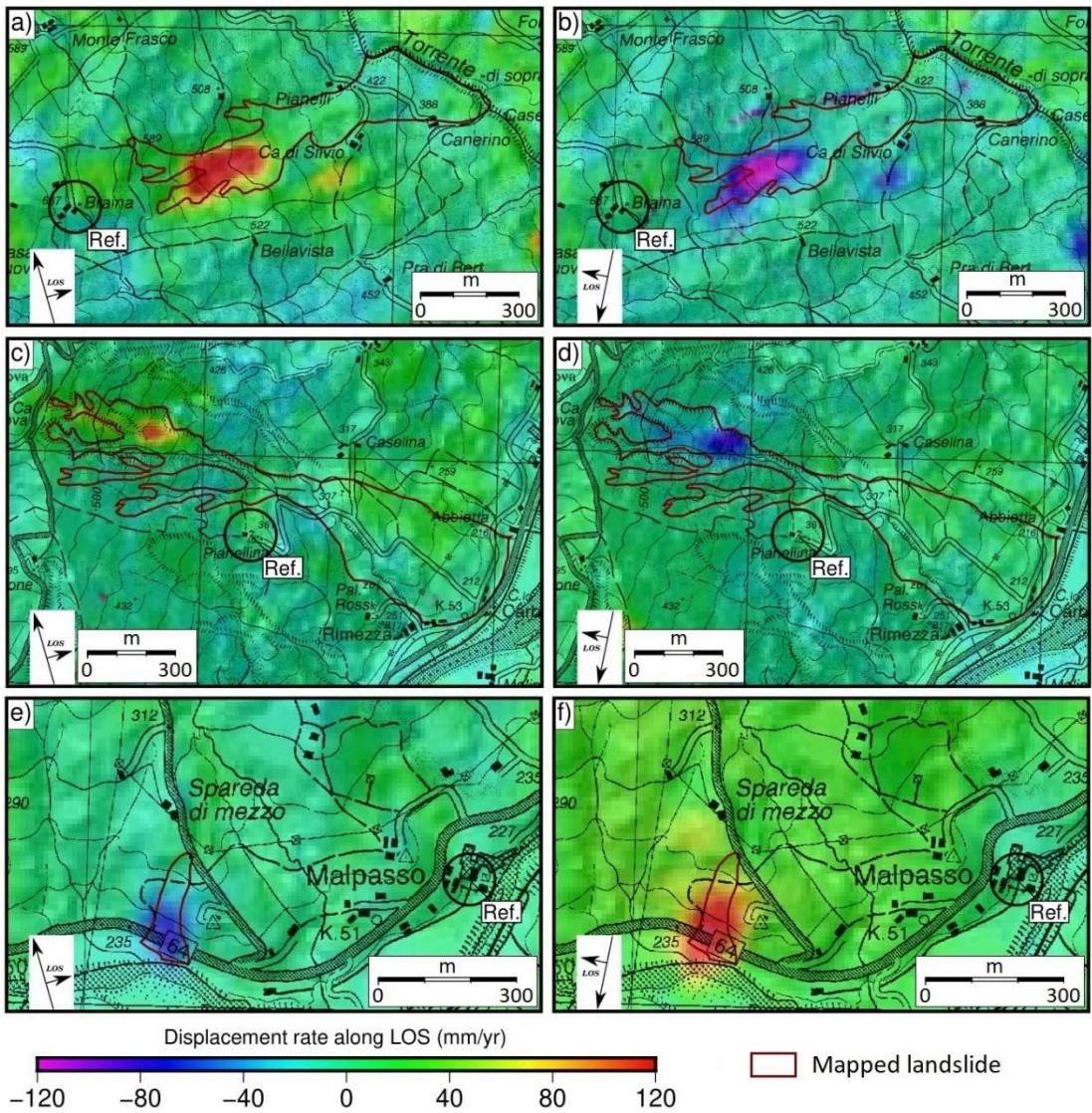

**Figure 6.** Results of site-specific analysis; stacks from January 2016 to December 2019 in both acquisition geometries for Braina (**a**,**b**), Carbona (**c**,**d**), Spareda (**e**,**f**).

Similarly, the Carbona earthflow underwent continuous deformations during the study period. Figure 6c,d shows that the upper portion of the main landslide body was affected by large displacements with maximum LOS-displacement values of 100–110 mm/year. The total stack in descending geometry (Figure 6d) gives values of the displacement rate along the LOS of 30 mm/year in the upper part (about 530 m a.s.l.); it also reveals that in the period of analysis, the whole upper portion of the main landslide body moved remarkably. The last pair of images (Figure 6e,f illustrate the Spareda landslide, showing the area affected by deformations during the four years preceding the landslide reactivation. In this case, the landslide is smaller, and the displacement signal entirely covers the mapped landslide. The movements extend from the toe of the landslide, locally incised by the Reno River at an altitude of 220 m a.s.l., up to 580 m a.s.l. The maximum displacement along the LOS exceeds 120 mm/year. The movements also affect the Municipal Road running along the slope, which was damaged by the landslide reactivation of November 2019. The descending orbit is almost parallel to the direction of movement of the Spareda landslide, while the ascending orbit forms an azimuth angle of 25–30°. Therefore, the descending dataset solely detects the vertical component of movement, while the ascending record a combination between vertical and horizontal. The

observed LOS-deformation signals are compatible with a downslope movement whose vertical lowering component can only be seen from the descending orbit. The ascending orbit, instead, records the vertical (range increase) and the horizontal components (range decrease) with opposite signs indicating that the latter is larger, as demonstrated by the negative LOS-displacement sign.

In order to investigate the dynamics of deformation through time, short timescale stacks can be used. We analyzed monthly stacks during the four year investigation period and discuss some of them below. Figure 7 shows the Carbona landslide's evolution in October and November 2019, the two months preceding the reactivation of the landslide, in ascending geometry.

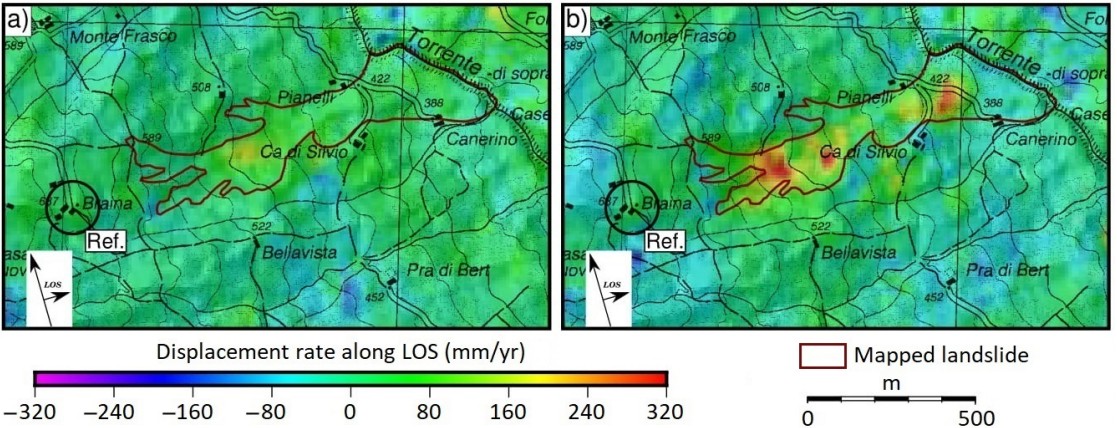

**Figure 7.** Monthly stacks were obtained for the period preceding the reactivation (15–21 November 2019). October (**a**), November (**b**).

During October (Figure 7a), the moving area is small and located between 480 m and 510 m a.s.l. The maximum displacement rates along the LOS reach about 250 mm/year, indicating an ongoing relevant acceleration when compared to typical velocities of dormant/inactive earthflow [31]. In November, the moving area is clearly larger and involves a large portion of the medium-upper earthflow body, at a short distance from few houses and a secondary road. LOS-displacement rates exceed locally 300 mm/year. The progressive acceleration that occurred in October and November 2019 preceded the reactivation of the earthflow that happened at the beginning of December. The corresponding monthly stacks capture the highest displacement rates within our analysis period. In the four years preceding the catastrophic failure, lower speeds were recorded, with minimum values during the summer and winter seasons and relatively higher values during spring and autumn. The average of all these speed values leads to the numerical values in Figure 6.

In the case of Spareda, we report three pairs of wrapped and unwrapped interferograms (Figure 8). The images are obtained by processing SAR data acquired from the ascending track 117. The temporal baseline of each interferogram is six days. The first pair (Figure 8a,b) precedes the failure, the second pair (Figure 8c,d) includes the reactivation date, and the last (Figure 8e,f) follows the failure. Cold colors and negative values indicate that displacements are towards the satellite, while warm colors and positive values describe displacements away from the satellite along the LOS.

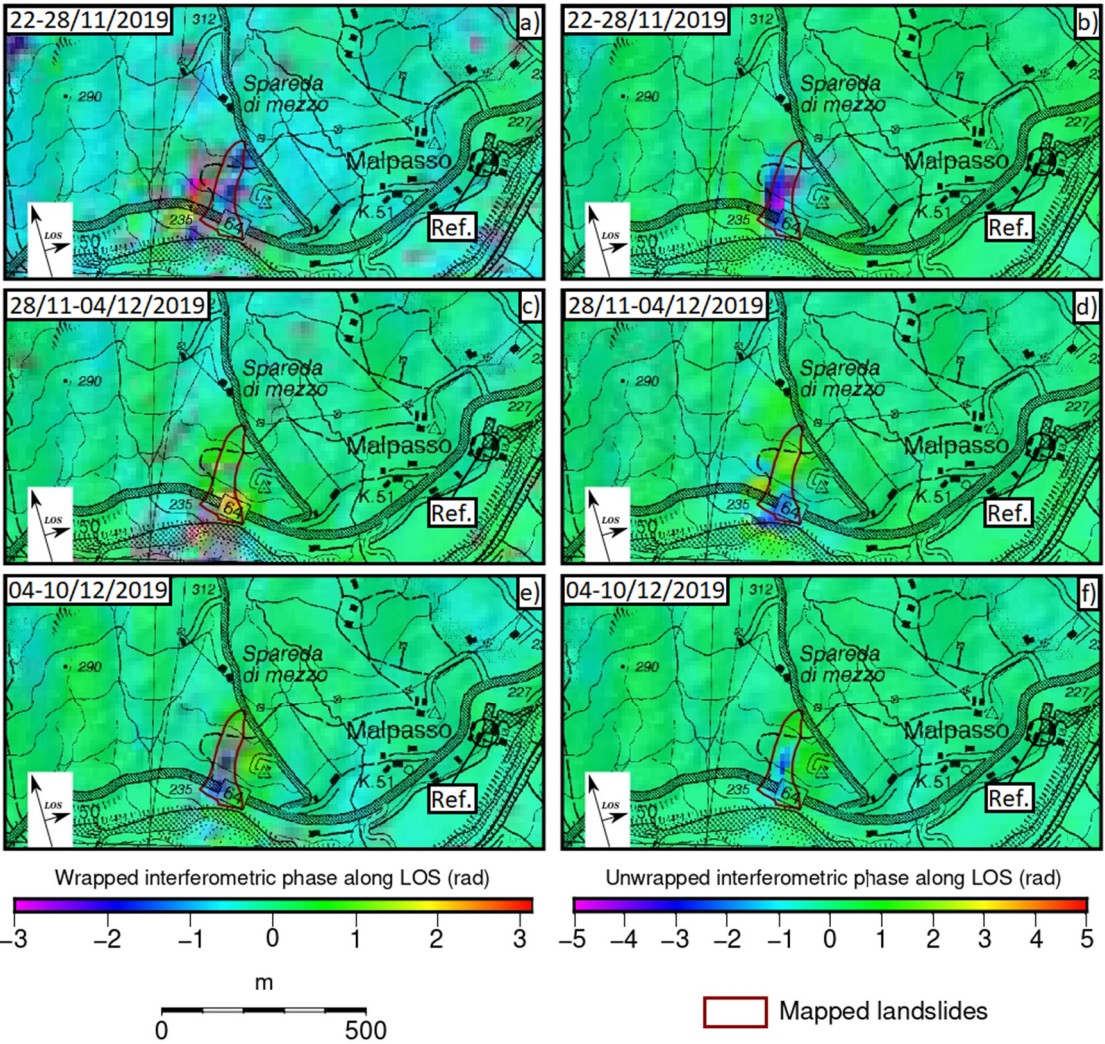

**Figure 8.** Six-day interferograms are covering the period of the Spareda landslide reactivation. Wrapped interferograms are reported in the left column (**a**,**c**,**e**) and corresponding unwrapped in the right column (**b**,**d**,**f**).

Figure 8a shows interferometric fringes that were correctly unwrapped into the range decrease in Figure 8b. Figure 8c is decorrelated on the landslide body because of the ongoing failure, and multiple phase jumps are visible ("fast-displacement decorrelation"). The phase ambiguities are only partially resolved, and it is possible that more than one cycle got lost, which is why the unwrapped displacements underestimate the true displacements (Figure 8d). It is worth noting that the range increase (movement away) from the satellite in the upper part could also be caused by a relevant vertical component due to the rotational kinematics of sliding. The last pair of images (Figure 8e,f) show again lower displacement rates with interferometric fringes that were correctly unwrapped. The signal indicates sustained sliding with displacements in directions away from the satellite in the upper part of the landslide body (surface lowering) and towards the satellite at the bulging toe.

*4.3. InSAR Analysis: Areal vs. Site-Specific (the Example of Spareda)*

In this section, we use the Spareda case study to describe the differences observed between areal and site-specific results. As an example, we considered three months in spring 2019. Figure 9 shows the comparison between the stacks obtained by the areal (a,b) and the site-specific InSAR analyses (c,d) for the ascending (a,c) and descending orbits (b,d).

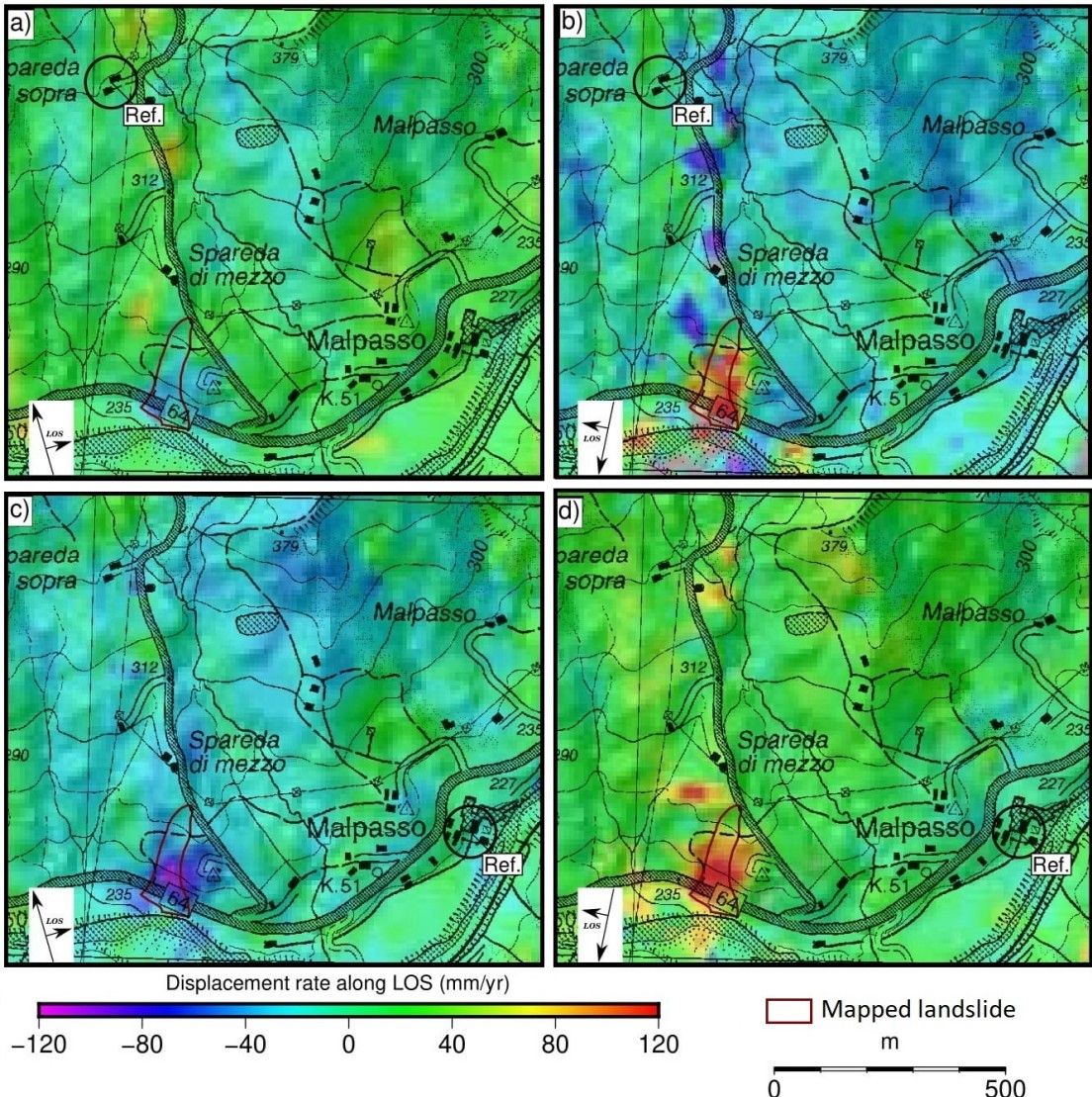

**Figure 9.** Comparison between three-month stacks obtained by the areal InSAR analysis (**a,b**) and site-specific InSAR analysis (**c,d**) for the ascending (**a,c**) and descending orbits (**b,d**).

The results indicate that the areal analysis can be used successfully to locate active landslides (Figure 9a,b), while the site-specific analysis improves the quality of results, both in terms of the spatial pattern of the signal as well as in terms of absolute velocities. The fact that the areal datasets contain higher residual noise can be explained by the different processing parameters. When working on a regional scale, compromises regarding the reference area but also regarding the interferograms included in the analyses must be made. It is, for example, possible that the overall coherence of an interferogram is good and will be included in the areal analysis. However, the same interferogram may be decorrelated on a specific landslide and would be excluded in the local-scale analysis.

During spring 2019, the areal analysis clearly reveals the deformation signal only in the descending dataset, but it provides lower quality data in the area outside the landslide; the displacement values along the LOS are also less accurate with this kind of analysis. The site-specific analysis is more accurate, showing less noisy results due to a more successful unwrapping process and of a careful site-specific interferogram selection.

### 4.4. Time Series vs. Precipitation Values

Site-specific results describe a complex behavior of sustained deformation occurring on the landslides in the four years preceding their reactivation. In order to describe

the temporal evolution of the three phenomena, we calculated the mean velocity of the deformation signal in the landslide for each monthly interferometric stack (i.e., July velocity is measured in the stack covering the period 1 to 31 July). The extraction polygon used to obtain the data reported in the time series (Figure 10) corresponds to the extension of the most intense deformation signal visible in Figure 6. It has an extension of 50,371 m$^2$ for Braina, 28,035 m$^2$ for Carbona and 8846 m$^2$ for Spareda. Figure 10 shows the time series obtained for the three landslides compared with precipitation data. The precipitation values are represented with the cumulated equivalent rain of 30 days that includes snow melting. The calculation is performed on a daily basis by summing the values pertaining to the preceding 30 days.

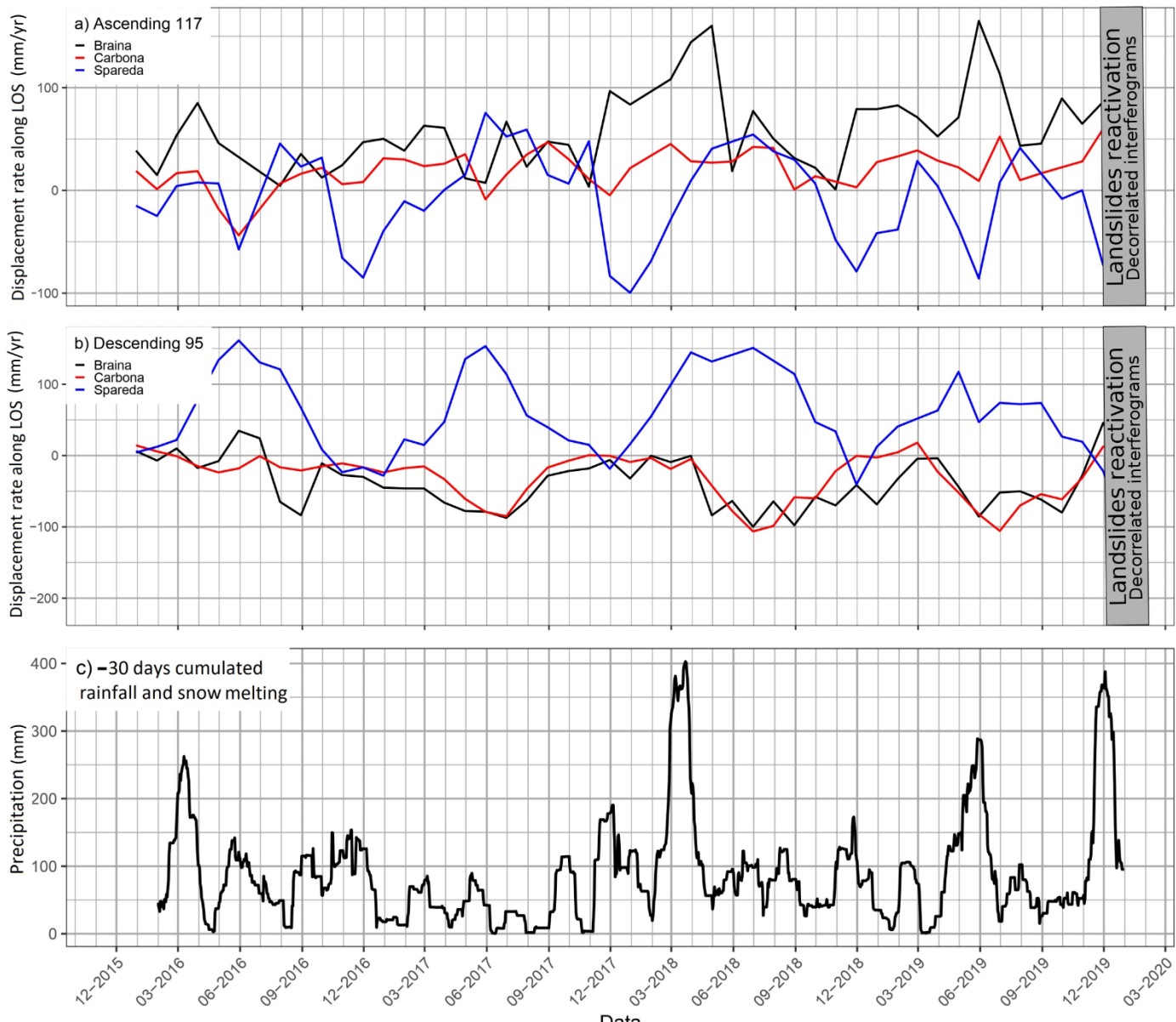

**Figure 10.** Time series of LOS (Line Of Sight) displacements rate for the ascending (**a**) and descending (**b**) orbits compared to 30 days cumulated precipitation calculated on a daily basis (**c**).

The major acceleration episodes are roughly synchronous for the three landslides and can be related to the wettest periods (March–April 2018, May and late 2019).

The comparison of the results obtained from the two orbits serves as a brief discussion. Ideally, in the case of gravitational slope movements, the displacement rate values mea-

sured by the ascending (looking eastward) and descending (looking westward) geometry should be opposite in sign when slopes have preferential east or west plunge and moderate steepness. The module would also be similar, provided the two geometries are symmetrical with respect to the direction of movement. Typically, however, the displacement signal is clearer in one of the two geometries. This is true for Braina and Carbona, where the ascending and descending LOS-displacement rates are, respectively, higher.

The main 30-day cumulative rainfall peaks occurred in March–April 2018, May 2019 and November–December 2019, the latter being the rainfall that triggered the reactivation of our case studies. Comparison of the first rainfall peak with the time series of displacement rates (Figure 10) suggests that Braina and Spareda experienced an acceleration during the precipitation, leading to displacement rate peaking occurring from April to May of the same year. The values exceed 100 mm/year and have an antithetic sign in the ascending and descending orbit. In the case of Carbona, a smaller acceleration is captured by the ascending dataset, but maximum displacement rates are relatively low (about 50 mm/year), and no corresponding acceleration is measured by the descending data.

Therefore, we do not interpret this information as completely reliable.

After the rainfall peak in May–June 2019, the deformation response has a similar pattern. In ascending orbit, we observe a clear acceleration in June for the Braina and Spareda case studies (Figure 10). Again, the response at Carbona is less clear and uncertain.

Using the analysis of individual interferograms as a complementary contribution, it seems that the peak of rainfall in late 2019 caused an increase in the displacement along the LOS, for both orbits, from the beginning of November. The descending results (Figure 10a red line) show a more impulsive response to the rainfall event. The acceleration that preceded the catastrophic failure in late 2019 is captured by the ascending dataset for all three cases. It shows an increase in the LOS displacement rate since November, i.e., since the onset of the rains.

For the Spareda landslide, the results obtained from ascending and descending acquisition are different because of the peculiar orientation of the satellite tracks compared to the direction of movement. Figure 10 shows phases where the antithesis between the two orbits is evident and others, such as December 2018, where such antithesis is not respected. In addition, in March 2018, there is a clear movement recorded in the descending dataset as a range increase, while in the ascending one, a decreasing LOS-velocity is observed. Such behavior probably results from the combination of vertical and horizontal movement associated with the prevailing downslope deformation (Figure 10a,b blue line). In this case, we interpret the data as the result of the relative orientation between the main direction of movement (i.e., SSW) and the satellite LOS. The observations are compatible with slope-parallel movement or with roto-translational sliding. This latter may contribute to additional vertical displacement whose effects are the only measured in descending geometry and that, instead, contribute to decrease the negative velocity measured in ascending geometry.

The second rainfall peak, in May 2019, shows a marked relationship with the time-series of the ascending orbit (Figure 10a). The acceleration starts with the rain and peaks in June. Later, during the same year, the rains of November caused a progressive acceleration of the movement until the landslide reactivation caused widespread decorrelation. This acceleration is more pronounced in the ascending dataset, which, in the case of Spareda, often appears more reliable compared to the descending.

## 5. Discussion and Conclusions

Our work is conceived following the reactivation of existing landslides in the Northern Apennines of Italy caused by the rains of autumn 2019. We selected three cases of reactivation and the area that includes them (about 60 km$^2$) to investigate the dynamics of gravitational slope movements by means of radar interferometry. Our aim is to detect and investigate active slope movements on an areal basis during the four years preceding the reactivations (2015–2019). Focused, site-specific analyses were then performed on the

three landslide cases to improve the quality and temporal resolution of the remotely sensed displacement information.

The areal mapping of deformation phenomena is performed using traditional two-pass differential interferometry, which helped to overcome the limitations that more rigorous multi-temporal approaches typically exhibit over largely vegetated areas where man-made structures or rock outcrops are sparse [55]. In our study, area good quality reflectors are rare, and only the large deep-seated landslides with small villages and houses on top can be assessed in detail via multitemporal InSAR approaches [15]. Traditional interferometry was used in the past over large areas taking advantage of long-wavelength (L-band) data from ALOS [14,22,56]. Due to the high acquisition frequency and small spatial baseline, C-band Sentinel data have also recently proved suitable for obtaining high-quality interferograms on landslides similar to the ones described in this paper [8,25].

From a technical point of view, our areal analysis results indicate that traditional InSAR can deliver almost continuous deformation maps of mountainous areas of the Northern Apennines characterized by a gentle slope (15–35°) and variable vegetation cover. Based on our experience, this latter prevents the detection of a coherent interferometric signal only when represented by dense forest cover. To detect sustained deformation through time that may correspond to active landslides, we use variable-duration stacked interferograms. This way, residual noise and artifacts are minimized, and displacement rates are averaged.

Our areal results show that stacked interferograms are useful to identify slope movements against surrounding stable slopes. During the 4 years of investigation, only 4 out of the 186 landslides mapped in the study area exhibited clearly recognizable deformation signals. Three of them correspond to the 2019 reactivations, and the fourth is represented by a landslide reactivated in March 2018. The analysis did not detect any noticeable and sustained movement neither on the remaining 182 mapped landslides nor along other slopes where first-time landslides may develop. Although the small number of moving landslides does not allow for statistical considerations, these results suggest that landslides prone to catastrophic reactivation are subject to measurable deformations for months or years before the actual failure occurs. In our case, failures were triggered by heavy and intense rainfall events whose occurrence cannot be predicted except in the short-term. However, detecting deformation signals using areal InSAR analysis may prove useful to identify active, slow-moving landslides more susceptible to reactivations than similar dormant cases. In the latter, the deformation could actually be absent or so slow that traditional InSAR (<3–4 cm per year) cannot detect it.

The areal InSAR analysis can also help to document the spatial evolution of the slope movements. For this purpose, the analysis must consider shorter interferometric stacks (seasonal to monthly) or even single interferograms. On a large scale, both exhibit an increasing presence of artifact or noise-related signals that make interpretation difficult. Therefore, we performed more focused analyses on the three active landslides reactivated at the end of 2019. The landslides can be classified as earthflows. They develop along gentle slopes made of chaotic clay shales and, like many others in our study area, exhibit a flow-like morphology associated with dominant sliding [31].

Site-specific analysis confirms that the earthflows were subject to noticeable deformation in the years preceding the reactivation. In the two larger cases (Braina and Carbona), ongoing deformation was concentrated in the upper part of the earthflow where multiple headscarps feed the main earthflow body, while in the smaller landslide (Spareda), the pre-failure deformation affected the whole body. In all cases, pre-failure deformation progressed with yearly averaged values of 60–90 mm/year and peak monthly values about twice.

During the reactivation phase, multiple phases jumps appear in the interferograms of the landslides. Given coherence is otherwise satisfactory, such a type of signal can be attributed to relatively fast displacements. However, it is difficult to correctly resolve the wrapped phase (Figure 9) even using the forward modeling of deformation [22–25]. We

believe that this is essentially due to the uneven deformation occurring on the surface that is favored by open crevices and fissures along, which high differential displacements occur. Despite unwrapping problems, during the November 2019 failure, interferograms show both surfaces lowering in the upper source area and bulging at the toe for the Spareda landslide (Figure 8). Such deformation pattern is compatible with a sliding style of movement, including some rotational components. In the two larger earthflows, instead, the reactivation interested only the upper source area and did not reach the accumulation zone.

Velocity variations during the pre-failure stage are well documented by the velocity time series that we measure monthly for each satellite track (Figure 10). Displacement rates range from virtually null values to more than 100 mm/month. The interpretation of velocity time series must consider the results derived from both flight geometries and the effects that the relative orientation between slope orientation and satellite LOS produce on the same results. We recognize that the quantitative meaning of displacement rates must be taken with great caution due to inherent uncertainties. However, the time-series show interesting qualitative trends. Late summer velocity decline is generally observed though the general trend cannot be described as seasonal but rather dependent on the rainfall regime. Repeated accelerations are generally observed during the wettest periods with delayed peaking compared to 30 days cumulated rainfalls. Interestingly, the reactivation of late 2019 occurred in coincidence with the peak of a rainfall period with no appreciable delay. Though capable of registering acceleration in the month preceding failure, our InSAR measures do not show any exceptionality compared to other accelerations, therefore, being of little use for the prediction of the time of failure. However, our analysis demonstrates that major precipitation events on a weekly to monthly scale are the main drivers of landslide accelerations and reactivations in our study area. More particularly, the catastrophic earthflow reactivations that occurred during the 4 year investigation period were caused by rainfalls and snow-melting episodes that exceeded 300 mm on a monthly basis. Unlike other authors, [14] we do not observe a clear trend of seasonal velocity variation, but rather, a general tendency to attain the lowest yearly velocities during late summer.

To conclude, we believe that traditional two-pass interferometry can be useful to investigate the dynamics of landslides. From a spatial perspective, it provided good coverage of the territory and proved capable of detecting active slope movements in the LOS-velocity range of few centimeters to few tens of centimeters per year. From an evolutionary perspective, it allows capturing the main landslide dynamics on a monthly, or possibly shorter, scale.

**Author Contributions:** Conceptualization, P.C. and A.S.; methodology, B.B., S.F. and A.S.; processing, P.C. and B.B.; validation, B.B., S.F. and M.B.; field investigation, P.C. and M.B.; writing—original draft preparation, P.C. and A.S.; writing—review and editing, P.C., B.B., S.F., M.B. and A.S.; supervision, A.S. All authors have read and agreed to the published version of the manuscript.

**Funding:** This research received no external funding.

**Institutional Review Board Statement:** Not applicable.

**Informed Consent Statement:** Not applicable.

**Data Availability Statement:** The data presented in this study are available on request from the corresponding author. The data are not publicly available due to their complex structure and storage limitations.

**Conflicts of Interest:** The authors declare no conflict of interest.

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
