# Peer review of "Deformation Detection in Cyclic Landslides Prior to Their Reactivation Using Two-Pass Satellite Interferometry"

_applsci, doi:10.3390/app11073156_

Round 1

Reviewer 1 Report

  1. 1: An Inset map showing the overview of the study area is needed.
  2. (As mentioned in Line 111) The three landslides (Braina, Carbona, and Spareda) can be better highlighted in Figure 1.
  3. Line 198: Citation number 443 > 43 (?) or 44 (?) Please verify!
  4. Line 196: Citation number 141 > 41 (?) or 14 (?) Please verify!
  5. Lines 216-219: That sentence needs to be rewritten. (You used the word of because 4 times)
  6. Lines 221-222: In the presence of … > This sentence needs to be rearranged.
  7. Lines 223-224: It is not very clear. Need to be rewritten and improved
  8. Line 227: like > such as
  9. Line 228: evolutions of it > its evolutions. I didn’t understand what you mean here by saying evolution. You probably want to highlight different PSI techniques based on various pixel selection methods and deformation models.
  10. Line 230: the word of “however” as a connecter is used wrongly to connect these two sentences and doesn’t show the contrasting idea clearly. Therefore, the whole paragraph should be rearranged accordingly
  11. Line 240: two ascending (tracks 14 and 117)
  12. Line 241: two descending (tracks 168 and 95)
  13. Line 241-242: The orbits used are the following: Track 15, Track 117 (ascending), track 168, and track 95 (descending).
  14. Lines 243-256: Can you please explain your selection criteria and rearrange the sentence again accordingly.
  15. Lines 255-256: due to effective landslide displacement: I didn’t understand this part of the sentence. Please clarify.
  16. Line 260: The visual inspection of 3000 interferograms does not seem realistic. Please explain if you used any systematic evaluation of interferogram noise and/or unwrapping errors.
  17. Lines 274-275: Authors should (?) > We discussed …. and interpreted
  18. Lines 274-275: Authors should … > This sentence needs to be rewritten.
  19. The findings and their implications should be (Why “should”?) discussed in the broadest context possible > Please improve the sentence.
  20. Lines 277-278: Future research directions may also be highlighted (This sentence belongs to the conclusion where you can mention the future perspective of the study)

Reviewer 2 Report

The issue of the possibility of predicting threats from landslide movements is very important. Traditional two-pass interferometry can be very useful to study the dynamics of landslides. This is also indicated by the results of previous observations of other researchers. Currently, however, it is important to create tools that allow for warning against the risk of landslide movement. The authors should look for relationships between the recorded active movements of the slopes and other parameters conducive to their activation. The case study presented should lead to more specific conclusions.

Reviewer 3 Report

  1. For the deformation detection of landslide, it’s crucial to know the depth of the sliding surface. I suggest that authors should make the geological profiles along the Barina, Carbona and Spareda landslides.
  2. Please give the detailed information for perpendicular and temporal baselines of SAR images used in this study. I suggest that authors can a new Table and a figure for this information.
  3. Section 4.2 InSAR site-specific analysis: It’s good to demonstrate the areal deformation maps by both ascending and descending orbits in LOS direction. However, it’s not easy to the readers to get the kinematic process of translational downslope movements. Thus, I suggest authors could transform the LOS direction deformation into downslope movements.
  4. Figures 6e and 6f: I am curious about the results of LOS-displacement along the Spareda landslide. Authors mentioned that Spareda is a south-west facing slope, for me, it’s almost a south-facing slope. Because the orbit of Sentinel-1 is subparallel to N-S direction, so it should be not sensitive to N-S horizonal movement. If the translational downslope movements do occur along the south-facing slope, only major vertical movements could be recorded by both ascending and descending orbits. These downslope vertical movements are away from the satellite (warm color), it’s unreasonable to get the opposite signals of ascending and descending orbits. This curiosity is also shown on Figures 8 and 9. Please explain it.
  5. Lines 452-462: Again, authors suggest a rotational component of the sliding may have produced vertical displacements in March 2018, however, if this hypothesis is true, that means both ascending and descending orbit will record the movement away from the satellite (positive value, warm color). I agree that a lower LOS-displacement due to the combination of vertical settlement and horizonal movement along the downslope in the ascending orbit. I suggest that authors should accurately calculate the contribution of horizontal movement along the downslope in the ascending orbit.

Minor comments:

  1. A regional locality map is necessary for the readers.
  2. Change mm/year to mm/yr.
  3. References #46 and #49 are identical.

Round 2

Reviewer 1 Report

Based on the revised manuscript, the paper still requires significant editing, as it is not written in sound English and cannot be accepted in its current form. As I reported in my first review, it is hard reviewing the scientific content of the manuscript with poor English, as the ideas are not clearly communicated and basic grammar mistakes and the general structure of the article is very distracting. Therefore, I reject the article.

Author Response

We extensively edited the manuscript in order to improve English language and style and to improve the overall clarity or the paper.

Reviewer 3 Report

The revised manuscript is now good enough for publication in Applied Sciences.

Author Response

We thank the reviewer for his comments.